# Structural Genomics of SARS-CoV-2 Indicates Evolutionary Conserved Functional Regions of Viral Proteins

**DOI:** 10.3390/v12040360

**Published:** 2020-03-25

**Authors:** Suhas Srinivasan, Hongzhu Cui, Ziyang Gao, Ming Liu, Senbao Lu, Winnie Mkandawire, Oleksandr Narykov, Mo Sun, Dmitry Korkin

**Affiliations:** 1Data Science Program, Worcester Polytechnic Institute, Worcester, MA 01609, USA; ssrinivasan@wpi.edu; 2Bioinformatics and Computational Biology Program, Worcester Polytechnic Institute, Worcester, MA 01609, USA; hcui2@wpi.edu (H.C.); zgao@wpi.edu (Z.G.); mliu5@wpi.edu (M.L.); slu3@wpi.edu (S.L.); wmkandawire@wpi.edu (W.M.); dkorkin@wpi.edu (D.K.); 3Computer Science Department, Worcester Polytechnic Institute, Worcester, MA 01609, USA; onarykov@wpi.edu

**Keywords:** 2019 novel coronavirus, SARS-CoV-2, 2019-nCoV, COVID-19, structural genomics, interactome, protein-protein interactions

## Abstract

During its first two and a half months, the recently emerged 2019 novel coronavirus, SARS-CoV-2, has already infected over one-hundred thousand people worldwide and has taken more than four thousand lives. However, the swiftly spreading virus also caused an unprecedentedly rapid response from the research community facing the unknown health challenge of potentially enormous proportions. Unfortunately, the experimental research to understand the molecular mechanisms behind the viral infection and to design a vaccine or antivirals is costly and takes months to develop. To expedite the advancement of our knowledge, we leveraged data about the related coronaviruses that is readily available in public databases and integrated these data into a single computational pipeline. As a result, we provide comprehensive structural genomics and interactomics roadmaps of SARS-CoV-2 and use this information to infer the possible functional differences and similarities with the related SARS coronavirus. All data are made publicly available to the research community.

## 1. Importance

Having infected over a hundred thousand people in more than a hundred countries across the globe, the new coronavirus, SARS-CoV-2, has a major health and economic impact worldwide. With a genome of the new virus sequenced, scientists learned that this virus is closely related to other coronaviruses, such as SARS-CoV and MERS-CoV. This discovery, however, leads to other questions. How different are the proteins of SARS-CoV-2, its main functional building blocks, from those of SARS-CoV? Does the new virus target the same proteins in human cells as the known coronaviruses? Most importantly, can the currently developed SARS or MERS antiviral drug candidates or other promising compounds be used to treat the new infection? This study lays out the first 3D roadmap of SARS-CoV-2 viral proteins and their functional complexes, enabling scientists to answer the above questions. The roadmap can be used to streamline the search for new antivirals and vaccines.

## 2. Introduction

Within two and a half months since its initial discovery in Wuhan, China, the novel deadly coronavirus SARS-CoV-2 has infected more than 100,000 people, with the death toll already surpassing that of the 2003 severe acute respiratory syndrome coronavirus (SARS-CoV) and 2012 Middle East respiratory syndrome coronavirus (MERS-CoV) outbreaks combined [1,2,3]. In spite of the instantaneous reaction by the scientific community and extensive worldwide efforts to address this health crisis, vaccines may be months and even years away [4,5]. For instance, a Phase I trial for a vaccine that treats SARS was announced in December 2004, two years after the disease outbreak [6]. Additionally, a vaccine against MERS, another infectious outbreak of the related coronavirus that emerged in 2012, was patented in 2019, with Phase I trials introduced in the same year [7,8]. Nevertheless, in the past two decades, a massive amount of work has been done to understand the molecular basis of the coronavirus evolution and infection, develop effective treatment in forms of both vaccines and antiviral drugs, and propose efficient measures for viral detection and prevention [9,10,11,12,13,14,15,16,17,18]. Structures of many individual proteins of SARS-CoV, MERS-CoV, and related coronaviruses, as well as their biological interactions with other viral and host proteins have been explored along with the experimental testing of the anti-viral properties of small-molecule inhibitors [16,19,20,21,22,23,24,25,26,27,28,29,30,31].

However, experimental investigation of the same scale for SARS-CoV-2 may take the research community years to obtain. Can it be facilitated? The answer lies in the use of the modern bioinformatics methods that can drastically streamline knowledge discovery by quickly providing important insights about possible molecular mechanisms behind infection, pinpointing likely protein targets for the anti-viral treatments, and predicting the efficacy of the existing antivirals developed for other coronaviruses. By leveraging previously known information on genome sequences as well as protein structures and functions, bioinformaticians have been successfully assisting virologists by structurally characterizing proteins of novel viruses, determining the evolutionary trajectories, identifying interactions with host proteins, and providing other important biological insights. In particular, a plethora of results has been achieved through comparative, or homology, modeling principles [32,33]. In addition to the global structural genomics, an initiative that focuses on determining the 3D structures of individual proteins on a genome scale [34], as well as to the specific efforts aimed at rapid structural characterization of proteins in emerging viruses [35,36,37,38], multiple works have used comparative modeling to predict the structures of protein–protein interaction complexes [39,40,41], facilitate structure-based drug discovery [33,42,43], infer protein functions [44], determine the macromolecular interaction network [45,46,47,48], and provide molecular insights into the viral evolution [49,50,51].

Here, using an integrated bioinformatics approach, we provide the first comprehensive structural genomics and interactomics analysis of the SARS-CoV-2. The structural information on the individual SARS-CoV-2 proteins and their interactions with each other and with human proteins allows us to accurately determine the putative functional sites. These functional sites, combined with an evolutionary sequence analysis of SARS-CoV-2 and the closely related human SARS-CoV and bat coronavirus proteomes, provide us with a structure-based perspective of the evolutionary diversity of SARS-CoV-2, and allow estimation of how similar the function of SARS-CoV-2 virus is when compared with SARS-CoV. Consequently, we can forecast how likely the antibodies and candidate ligands that are efficient in inhibiting the SARS-CoV functions will be efficient in doing the same for SARS-CoV-2.

## 3. Materials and Methods

The goal of this work is to identify the evolutionary differences between SARS-CoV-2 and the closest coronavirus species, human SARS-CoV and bat SARS-like coronavirus, and to predict their possible functional implications. To do so, we structurally characterized individual proteins as well as intra-viral and human–virus protein complexes, extracted the information on their interaction interfaces and ligand binding, and superposed the evolutionary difference and conservation information with the binding information. Specifically, our integrative computational pipeline included the following five steps. First, for each of the candidate SARS-CoV-2 proteins a set of sequentially similar coronavirus proteins was determined and aligned. Second, structural models of SARS-CoV-2 proteins were obtained using template-based, or homology, modeling. Third, the protein–protein interaction complex structures of SARS-CoV-2 proteins interacting with each other and/or with the human proteins were determined using a multi-chain comparative modeling protocol. Fourth, the protein-binding sites were extracted from the obtained models of protein–protein interaction complexes, and protein–ligand binding sites were extracted from the evolutionarily close coronavirus protein–ligand templates and mapped to the relevant structural models of SARS-CoV-2 through structural alignment. Fifth, the information on the evolutionary differences and conservations between the protein sequences of SARS-CoV-2 and the related coronaviruses were extracted from the above protein sequence alignments and mapped onto the structural models of the SARS-CoV-2 proteins to determine if the protein- and ligand-binding sites were functionally conserved. Lastly, a joint human–virus and virus–virus interactome was constructed through homology and further analyzed. All the obtained models and sequence alignments were made publicly available to the research community.

### 3.1. Protein Sequence Data Collection and Analysis

Available sequences for protein candidates wS, wORF3a, wE, wM, wORF6, wORF7a, wORF7b, wORF8, wN, and wORF10 were extracted from the NCBI Virus repository [52] (collected on 29 January 2019) and then used in the sequence analysis and structural modeling (see Appendix A containing sequence alignment files of each protein and the functional mappings of protein and ligand binding sites). The Uniprot BLAST-based search was performed for each of the proteins using default parameters. From the results of each search, the final selection was done based on the pairwise sequence identity (>60%) as well as the evolutionary relationship (*Coronaviridae* family). Each of SARS-CoV-2 proteins was then aligned with the corresponding coronavirus proteins using a multiple sequence alignment method Clustal Omega (EMBL-EBI, Cambridge, UK) [53].

### 3.2. Structural Characterization of Protein and Protein Complexes

The structure of each protein was determined using a single-template comparative modeling protocols with the MODELLER software (UCSF, CA, USA) [54]. First, the template for each protein sequence was identified using a PSI-BLAST search in the Protein Data Bank (PDB) (Research Collaboratory for Structural Bioinformatics) [55]. In general, a structural template with the highest sequence identity was selected out of those that covered at least 50 residues of the target sequence with at least 30% sequence identity. The polyprotein wORF1ab was first split into 16 putative proteins based on its alignment with the human SARS-CoV polyprotein, with each protein then independently searched against PDB. In total, structural templates for 17 proteins were chosen (Table 1, Figure 1, Figure 2 and Figure 3). In some cases, several independent templates, each covering an individual protein domain of a target SARS-CoV-2 protein, were selected. The obtained template was used in the comparative modeling protocol, generating five models. Each model was assessed using the DOPE statistical potential [56]; the best-scoring model was selected as a final prediction.

To model a protein–protein interaction complex, a multi-chain modeling protocol was used [39]. Specifically, we aligned the corresponding pairs of homologous proteins and combined them into a single alignment where the individual chains were separated by “/” symbol. The alignment was used as an input together with the multi-chain structural template of a homologous complex. In the case of a viral–human interaction (these were the only virus–host structural templates found), the human protein remained the same in the alignment. In total, 18 structural templates of protein complexes involving homologs of 14 SARS-CoV-2 proteins were retrieved (Figure 3). Similar to the single-protein protocol, five candidate models were generated for each complex conformation, and each model was assessed using the DOPE statistical potential, following by selection of the best-scoring model as a final prediction.

### 3.3. Mapping of Functional Regions and Evolutionary Conservation 

We next extracted protein- and ligand-binding sites and mapped them onto the models of SARS-CoV-2 proteins. For protein binding sites, the obtained modeled structures of protein complexes that involved SARS-CoV-2 proteins were considered. For each SARS-CoV-2 protein in a protein–protein interaction complex, we identified all binding residues that constituted its protein-binding site. Given an interaction between two proteins, a residue on one protein was defined as a protein binding site residue if there was at least one pair of atoms, one from this residue and another from a residue in the second protein, with Van der Waals (VDW) surfaces not farther than 1.0 Å from each other. Using this definition, the binding sites were identified with UCSF Chimera (UCSF, CA, USA) [57]. The ligand binding sites were identified and mapped using a different protocol—we relied on the default definition of protein–ligand binding site residues from PDB 3D Ligand View, since this standard was widely accepted. Once all ligand binding site residues were identified for a protein from a related coronavirus, the residues were mapped onto the surface of the corresponding SARS-CoV-2 protein, guided by a structural alignment between the two proteins, which put residues from both proteins into a one-to-one correspondence.

### 3.4. Inferring Intra-Viral and Virus–Host Protein–Protein Interaction Networks 

Next, we leveraged the obtained protein homology information to predict and map all possible intra-viral and virus–host protein interactions at the systems level. Since the SARS-CoV-2 genome exhibited substantial similarity to the 2002 SARS-CoV genome [58] and proteome [59], we hypothesized that many of the interactions observed in the SARS-CoV proteome would be preserved in the SARS-CoV-2 proteome too, unless the corresponding binding sites were affected. The information in the interactome would help one understand the global mechanistic processes of the viral molecular machinery during the viral infection, survival within the host, and replication. With this knowledge, one could discern the protein interactions that were crucial for transmission and replication. These interactions could then be potential candidates for inhibitory drugs [60]. Furthermore, using the network biology approach, one could identify hubs and bottlenecks, new to SARS-CoV-2, that could again be targeted by the antiviral drugs.

For this purpose, we created a comprehensive integrated SARS-CoV interactome that consisted of both intra-viral and virus–host interactions. The SARS-CoV intra-viral interactome was created using the published data where the SARS-CoV ORFeome was cloned, and a genome-wide analysis of viral protein interactions was performed through yeast-two-hybrid (Y2H) matrix screens [61]. The Y2H matrix screen was summarized by combining interactions in one direction, both directions and self-interactions. We also included intra-viral interactions gathered from the literature review [61]. The aggregated intra-viral interaction network consisted of 31 proteins and 86 unique interactions. 

We then constructed the SARS-CoV–Host interactome through the published Y2H interaction data [62]. We also included virus–host interactions mined from a literature survey of 5000 abstracts [62]. The curated virus-host interaction network consisted of 118 proteins, including 93 host proteins, and 114 unique virus-host interactions. Next, to create an integrated network, the two individual networks were imported in Cytoscape (Cytoscape Consortium) [63] and merged to form a unified interactome, representing both intra-viral and virus-host interactions. Finally, we included our predictions from structural modeling of SARS-CoV-2 intra-viral and virus–host interactions, surveyed recent literature on predicted SARS-CoV-2 interactions and annotated them accordingly in the unified interactome. The unified interactome consisted of 125 proteins (94 host proteins) and 200 unique interactions. In addition, the networks were pruned for duplicate edges, and a network topology analysis was performed to compute a set of summary statistics (degree distribution, clustering coefficient, and other important characteristics) that characterized the three networks. Finally, the 200 putative interactions were annotated based on the extent to which the protein binding sites of the SARS-CoV-2 proteins were altered, compared to their SARS-CoV homologs. Based on the evolutionary conservation analysis of the putative protein binding sites extracted from the modeled complexes, some interactions were annotated as potentially disrupted.

## 4. Results

The recently sequenced genomes of SARS-CoV-2 strains combined with the comparative analysis of the SARS-CoV genome organization and transcription allowed us to construct a tentative list of gene products [64]. It was suggested that SARS-CoV-2 had 16 predicted non-structural proteins (referred to as wNsp1–wNsp16 here) constituting a polyprotein (wORF1ab), followed by (at least) 13 downstream open reading frames (ORFs): Surface glycoprotein (or Spike), ORF3a, ORF3b, Envelope, Membrane, ORF6, ORF7a, ORF7b, ORF8, Nucleocapsid, ORF9a, ORF9b, and ORF10, which we refer in this work to as wS, wORF3a, wORF3b, wE, wM, wORF6, wORF7a, wORF7b, wORF8, wN, wORF9a, wORF9b, and wORF10, respectively. The three viral species whose proteins shared the highest similarity were consistently the same: human SARS coronavirus (SARS-CoV), bat coronavirus (BtCoV), as well as another bat betacoronavirus (BtRf-BetaCoV).

### 4.1. Comparative Analysis of SARS-CoV-2 Proteins With the Evolutionary Related Coronavirus Proteins Reveals Unevenly Distributed Large Genomic Insertions 

Searching against UniProt database (UniProt Consortium) [65] resulted in matches for the polyprotein (wORF1ab), all four structural proteins (wS, wE, wM, and wN), and six ORFs (wORF3a, wORF6, wORF7a, wORF7b, wORF8, and wORF10), also referred to as the accessory proteins. The closest protein matches from UniProt shared sequence identity with the related SARS-CoV-2 proteins as high as 91% (with wORF1ab and wN) and as low as 57% (with wORF8) (Appendix A). The majority of differences were single-residue substitutions spread across the protein sequence (see multiple sequence alignment files for all proteins in Appendix A).

Perhaps the most profound differences lie in the sequences of the multi-domain protein wNsp3 and surface protein wS: our analysis revealed that, compared to related coronavirus proteins, the two proteins had large sequence inserts (multiple sequence alignments for both proteins can be found in Appendix A, files wORF1AB_matches.aln and wS_matches.aln). In particular, wNsp3 had a novel large (25–41 res., region 983–1023 of wORF1ab, depending on the alignment method) insert between its two putative functional domains, which are homologous to the N-terminal domain and adenosine diphosphate ribose 1″ phosphatase (ADRP, recently renamed to Macrodomain-1 or Mac-1) of SARS-CoV [66,67] (Appendix A, file wORF1AB_matches.aln). Interestingly, the closest matching peptide was found in C-Jun-amino-terminal kinase-interacting protein 4 (seq. identity is 46%) of *Labrus bergylta*, a species of marine ray finned fish. Being significantly more diverse than the other three structural proteins, wS was found to have 4 inserts (4–6 res.) that seemed unique to SARS-CoV-2 and two additional inserts shared with human SARS-CoV proteins (Appendix A, file wS_matches.aln).

### 4.2. Three Recent Strains of Bat SARS-like Coronavirus from 2013, 2015 and 2017 Share Extremely High Proteome Similarity with SARS-CoV-2 

The unusually low conservation of ORF6, ORF8, and surface proteins between SARS-CoV-2 and human SARS-CoV, bat coronavirus BtCoV, as well as another bat betacoronavirus, BtRf-BetaCoV, prompted us to perform an expanded search for ORF8 homologs using the blatp tool of NCBI BLAST against a large non-redundant protein sequence repository (nr) [68]. Our search resulted in three new homologs of ORF8 not reported in UniProt that were originated from three different isolates of bat SARS-like coronavirus: bat-SL-CoVZC45 (GenBank ID: MG772933, collected in 2017), bat-SL-CoVZXC21 (GenBank ID: MG772934, collected in 2015), and RaTG13 (GenBank ID: MN996532, collected in 2013). The protein sequences from these isolates shared a striking similarity with wORF8, unseen in other strains before: the sequence identities between each of these three homologs and wORF8 ranged between 94% and 95%. Further analysis showed that the proteomes of these isolates shared even higher sequence identity with the other proteins of SARS-CoV-2: from 88.4% to 100% for 2015 and 2017 isolates, and even higher, 97.4%–100% for the 2013 isolate (Appendix A).

In spite of the significant similarity of the three isolates to SARS-CoV-2, important differences were observed. First, similar to other viruses, the 2017 and 2015 isolates did not have the four sequence inserts that were found in wS. Second, neither of the two isolates had the large insert between the two domains of wNsp3 from wORF1ab described above. On the contrary, the 2013 isolate had both, the four sequence inserts in its surface protein, matching those in wS, and the large insert in Nsp3, although the sequence of the large insert is different from that one in wNsp3.

### 4.3. Structural Genomics and Interactomics Analysis of SARS-CoV-2

Next, as a result of a comprehensive comparative modeling effort, we were able to structurally characterize 17 individual proteins, including 13 non-structural proteins of wORF1ab (wNsp1, wNsp3, wNsp4, wNsp5, wNsp7, wNsp8, wNsp9, wNsp10, wNsp12, wNsp14, wNsp15, wNsp16), three structural proteins (wE, wN, and wS), as well as one ORF (wORF7a). For two proteins, wNsp3 and wN, multiple individual domains were modeled (Figure 1, Figure 2). The templates for the majority of the models were the homologous protein structures from other coronaviruses, with a high target-to-template sequence similarity (seq. ids: 75–96%, except for ORF8). N-terminal and C-terminal domains of wN corresponded to N-terminal RNA-binding domain and C-terminal dimerization domain of SARS-CoV, respectively. The modeled domains 1–6 of wNsp3 corresponded to (1) N-terminal domain; (2) Mac-1; (3) SUD domain (SARS-specific unique domain) containing two macrodomains; (4) SUD domain C; (5) the papain-like protease PLPro domain; and (6) Y domain of SARS-CoV (Figure 1). A previously identified transmembrane domain of SARS Nsp3 was mapped to the sequence of wNsp3, but could not be modeled due to the lack of a template structure.

The structural analysis of the modeled proteins combined with the sequence conservation analysis revealed several findings. First, we found that the mutated residues tended to locate on the protein’s surface, supporting previous observations in other families of RNA viruses that the core residues of viral proteins were more conserved than the surface residues [49,69,70]. Furthermore, in a substantial number of proteins, distributions of mutated positions exhibited spatial patterns, with groups of mutations found to form clusters on the protein surfaces (Figure 2). These obtained models were then used as the reference structures to map and analyze the protein-binding and ligand-binding sites.

Next, using comparative modeling, we structurally characterized protein interaction complexes for both intra-viral interactions (homo- and hetero-oligomers) and host–viral interactions, where the host proteins were exclusively human. In total, we obtained structural models for 16 homo-oligomeric complexes, three hetero-oligomeric complexes, and eight human–virus interaction complexes including multiple conformations (Figure 1 and Figure 3). The intra-viral hetero-oligomeric complexes included exclusively the interactions between the non-structural proteins (wNsp7, wNsp8, wNsp10, wNsp12, wNsp14, and wNsp16). The modeled host–viral interaction complexes included three types of interactions: non-structural protein wNsp3 (papain-like protease, PLpro, domain) interacting with human ubiquitin-aldehyde, surface protein wS (in its trimeric form) interacting with human receptor angiotensin-converting enzyme 2 (ACE2) in different conformations, as well as the same protein wS interacting with several neutralizing antibodies. Based on the obtained models, the protein interaction binding sites were extracted and analyzed with respect to their evolutionary conservation.

### 4.4. Evolutionary Conservation and Divergence of Functional Regions of SARS-CoV-2

The analysis of the evolutionary conservation of protein binding sites revealed several patterns. First, we found that all protein binding sites of non-structural proteins involving in the intra-viral heteromeric complexes, wNSP7-wNsp8-wNsp12, wNsp10-wNsp14, and wNsp10-wNsp16, were either fully conserved or allowed at most one mutation on the periphery of the binding region, in spite of the fact that each protein had multiple mutations on its surface (Figure 4A, and file wORF1AB_Nsp7_Nsp8_Nsp10_Nsp12_Nsp14_Nsp16_PBS_mapped.pdf in Appendix A). Furthermore, we observed the same behavior when analyzing the interaction between the papain-like protease PLpro domain of wNsp3 and human ubiquitin-aldehyde (Figure 4B, and file wORF1AB_wNsp3_domain5_human_PBS_mapped.pdf in Appendix A): the only two mutated residues were located on the border of the binding region and thus were unlikely to disrupt the protein–protein interaction.

Protein wS presented a striking contrast to the majority of SARS-CoV-2 proteins due to its heavily mutated surface (Figure 5, Appendix A: wS_matches.aln and wS_human_PBS_mapped.pdf in Appendix A). First, the four novel sequence inserts and two inserts shared with the closest strains were expected to affect the protein’s function. Interestingly, three of the novel inserts were located in the first NTD domain, while the fourth one was located immediately before the S2 cleavage site and inside the homo-trimerization interaction interface. While the RBD domain of wS was not affected by those inserts, it was the most heavily mutated region of wS with potentially altering functional effects on the interactions with the human ACE2 receptor and monoclonal antibodies mAb 396 [71] and mAb 80R [72] (Figure 4B). The NTD domain had been considered a target of another antibody, Ab G2, previously shown to work in MERS-CoV [73]. However, based on the structural superposition of the NTD domain, it seemed likely that the potential interaction would be disrupted by the novel sequence inserts to wS.

Finally, the analysis of seven ligand binding sites (LBS) for multiple candidate inhibitors previously identified, with their interactions structurally resolved, for four proteins in SARS-CoV and MERS-CoV, showed that many of the LBS were intact in the corresponding SARS-CoV-2 proteins, such as wN, wNsp3, wNsp5, and wNsp16 (Figure 4 and Appendix A: wN_LBS_mapped.pdf, wNsp3-domain5_LBS_mapped.pdf, wNsp5-nonSARS_LBS_mapped.pdf, wNsp5-SARS_LBS1_mapped.pdf, wNsp5-SARS_LBS2_mapped.pdf, wNsp14_LBS_mapped.pdf, and wNsp16_LBS_mapped.pdf). For wNsp14, the LBS for several inhibitors were mutated, while a co-localized binding site for another ligand was intact.

### 4.5. Joint Intra-Viral and Human–Virus Protein–Protein Interaction Network for SARS-CoV Indicates Potential System-Wide Roles of SARS-CoV-2 Proteins

After constructing the individual networks of intra-viral interactions and virus–host interactions, they were merged to form a unified network of SARS-CoV interactions, with the hypothesis that most interactions would be conserved in SARS-CoV-2 (Figure 6). The network analysis of all three networks showed that unifying the intra-viral and virus–host networks reduced the number of disconnected components (islands) in the SARS-CoV–Host interactome (Table 2). The clustering coefficient and average node degree were both higher in the unified interactome compared to the virus–host interactome.

The network topology of the unified interactome indicated the presence of several viral hubs for the structural, non-structural, and accessory proteins (Figure 6). The viral proteins exclusively formed all the hubs, with a few of specific interest. ORF9b was one of the largest hubs in the network thought to play a secondary role only in intra-viral interactions and not necessary for replication [61]. However, a recent study showed that ORF9b hindered immunity by targeting the mitochondria and limited the host cell responses [74]. Nsp8 was the other major hub, which interacted with other replicase proteins, including two other hubs, Nsp7 and Nsp12, which together played a crucial role in replication [75].

Another crucial non-structural protein was Nsp1, also present in the SARS-CoV-2, which was known to be an inhibitor of the innate immune response. The host interaction partners of NSP1 modulated the Calcineurin/NFAT pathway that played an important role in immune cell activation [62]. Overexpression of Nsp1 was associated with immunopathogenecity and long-term cytokine dysregulation as observed in severe SARS cases. Additionally, inhibition of cyclophilins/immunopilins (host interaction partners) by cyclosporine A (CspA) blocked the replication of CoVs of all genera.

There was an important interaction between the SARS-CoV spike protein (S) and the host angiotensin-converting enzyme 2 (ACE2), as it was associated with cross-species and human-to-human transmissions. The same interaction was also inferred from structural modeling between SARS-CoV-2 wS and ACE2. Surprisingly, it was recently established that the interaction was preserved [76], in spite of a number of mutations in the receptor binding site of wS. Similar to SARS-CoV [77], wS protein in SARS-CoV-2 was expected to interact with type II transmembrane protease (TMPRSS2) [78] and was likely to be involved in the inhibition of antibody-mediated neutralization. Thus, wS remained an important target for vaccines and drugs previously evaluated against SARS and MERS, while a neutralizing antibody targeting the wS protein could provide passive immunity [79]. In addition, there were seven interactions from SARS-CoV determined by structural characterization of the protein complexes that were predicted to be either conserved or potentially disrupted in SARS-CoV-2 (green edges in Figure 6). An important target for vaccines was the surface (S) protein which was evaluated in SARS-CoV and MERS-CoV, with the idea that a neutralizing antibody targeting wS protein could provide passive immunity for SARS-CoV-2 [79]. We also structurally modeled interactions between the SARS-CoV-2 wS protein and three human monoclonal antibodies that were previously studied in SARS-CoV for immunotherapy and mapped the information about the evolutionary conserved and diverse surface residues. We found that two interactions of wS, with mAb 396 [71] and mAb 80R [72], were likely to be disrupted due to the heavily mutated binding sites while another interaction, with Ab G2 [73], was likely to be disrupted due to a novel insert into the wS sequence. Together, these findings may provide structure-informed guidelines in the search for potential antibody treatment.

## 5. Discussion

This work provides an initial large-scale structural genomics and interactomics effort towards understanding the structure, function, and evolution of the SARS-CoV-2 virus. The goal of this computational work is two-fold. First, by making the structural road map and the related findings fully available to the research community, we aim to facilitate the process of structure-guided research where accurate structural models of proteins and their interaction complexes already exist. Second, by providing a comparative analysis between the new virus and its closest relatives from the perspective of protein- and ligand-binding, we hope to help experimental scientists in their deciphering of the molecular mechanisms implicated in infection by the new coronavirus as well as in vaccine development and antiviral drug discovery. 

Through integrating the information on structure, function, and evolution in a comparative study of SARS-CoV-2 and the closely related coronaviruses, one can make several preliminary conclusions. First, the extended peptide sequence newly introduced to wNsp3 between two structurally and possibly functionally independent domains of this protein, might act as a long inter-domain linker, thus extending the conformational flexibility of this multi-domain protein. Second, the presence of the four novel inserts and one highly variable region of the surface protein wS and the analysis of this large-scale sequence change with respect to intra-viral and viral–host interactions lead us to conclude that these inserts might have structural impact on the homo-trimeric form of the protein as well as impact the functions carried out by the NTD domain. Third, the structurally modellable repertoire of the SARS-CoV-2 proteome also points to the interesting targets for structural biology and hybrid methods. For instance, the whole structures of the multi-domain proteins wN and wNsp3 could be resolved by integrating comparative models of the individual domains with the lower-resolution techniques that cover the entire protein, such as cryogenic electron microscopy (CryoEM). The structure of wNsp3 is especially interesting because of the presence of the novel peptide introduced between the two structural domains of the protein.

The evolutionary analysis of the protein- and ligand-binding sites mapped on the surfaces of SARS-CoV-2 may provide new insights into the virus functioning and its future treatment. The 100% or near 100% evolutionary conservation of the protein binding sites on the surfaces of non-structural proteins wNsp7, wNsp8, wNsp10, wNsp12, wNsp14, and wNsp16 that correspond to the intra-viral interactions for three complexes is consistent with our previous observations that the intra-viral interactions are significantly more conserved than viral–host interactions [49,50,80]. However, the near-perfect conservation of the human ubiquitin-aldehyde protein binding site on the surface of wNsp3 is rather intriguing, suggesting the critical role of this interaction in the functioning of SARS-like coronaviruses. Lastly, the conservation of the ligand-binding sites for many putative inhibitors previously developed against SARS and MERS suggests the development of antiviral drugs as a promising direction for addressing this new global health threat.

## Figures and Tables

**Figure 1 viruses-12-00360-f001:**
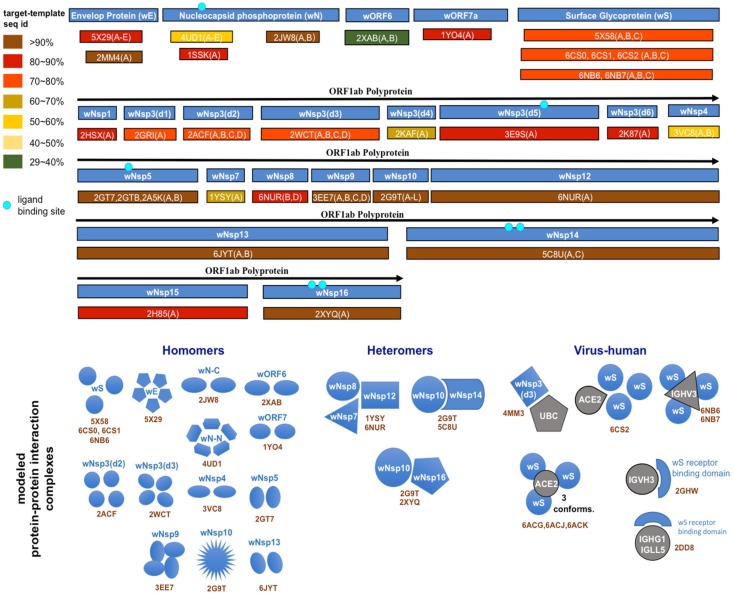
Structural genomics and interactomics road map. Shown are the individual proteins and protein complexes targeted for structural characterization together with PDB ID of their templates from the Protein Data Bank (PDB).

**Figure 2 viruses-12-00360-f002:**
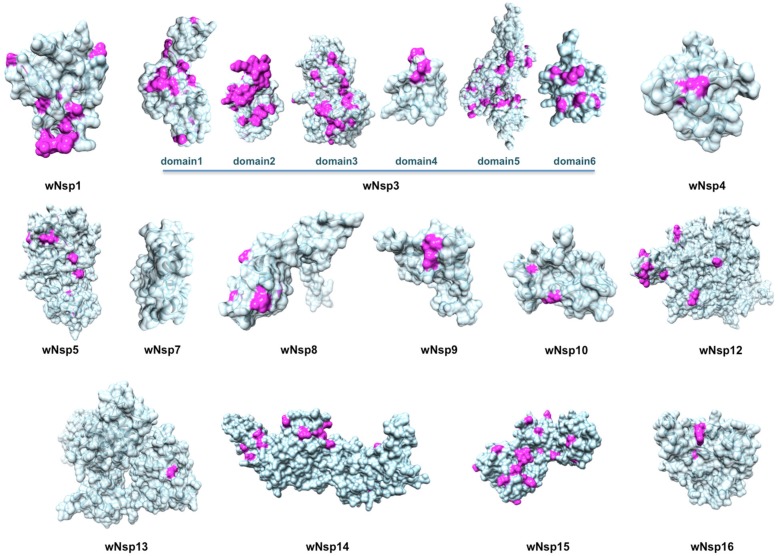
Structurally characterized non-structural proteins of SARS-CoV-2. Highlighted in pink are mutations found when aligning the proteins against their homologs from the closest related coronaviruses: human SARS-CoV, bat coronavirus BtCoV, and another bat betacoronavirus BtRf-BetaCoV. The structurally resolved part of wNsp7 shares 100% sequence identity to its homolog.

**Figure 3 viruses-12-00360-f003:**
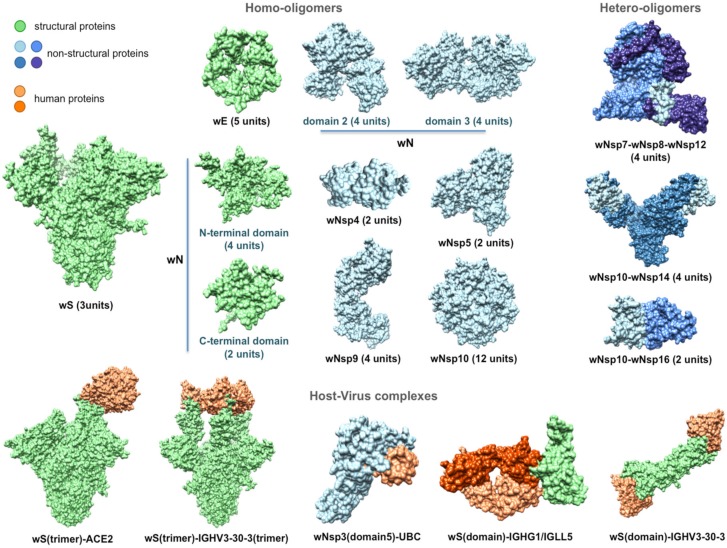
Structurally characterized intra-viral and host–viral protein–protein interaction complexes of SARS-CoV-2. Human proteins (colored in orange) are identified through their gene names. For each intra-viral structure, the number of subunits involved in the interaction is specified.

**Figure 4 viruses-12-00360-f004:**
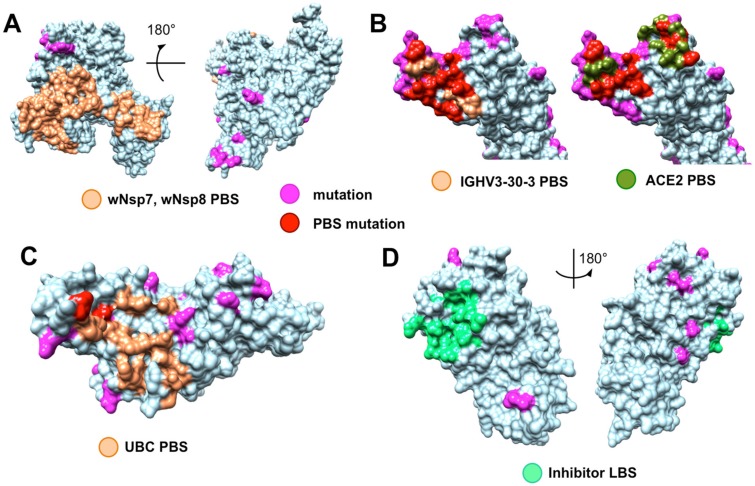
Evolutionary conservation of functional sites in SARS-CoV-2 proteins. (**A**) Fully conserved protein binding sites (PBS, light orange) of wNsp12 in its interaction with wNsp7 and wNsp8, while other parts of the protein surface shows mutations (magenta); (**B**) Both the major monoclonal antibody binding site (light orange) and the ACE2 receptor binding site (dark green) of wS are heavily mutated (binding site mutations are shown in red) compared to the same binding sites in other coronaviruses; mutations not located on the two binding sites are shown in magenta; (**C**) Nearly intact protein binding site (light orange) of wNsp (papain-like protease PLpro domain) for its putative interaction with human ubiquitin-aldehyde (binding site mutations of the only two residues are shown in red, non-binding site mutations are shown in magenta); (**D**) Fully conserved inhibitor ligand binding site (LBS, green) for wNsp5; non-binding site mutations are shown in magenta.

**Figure 5 viruses-12-00360-f005:**
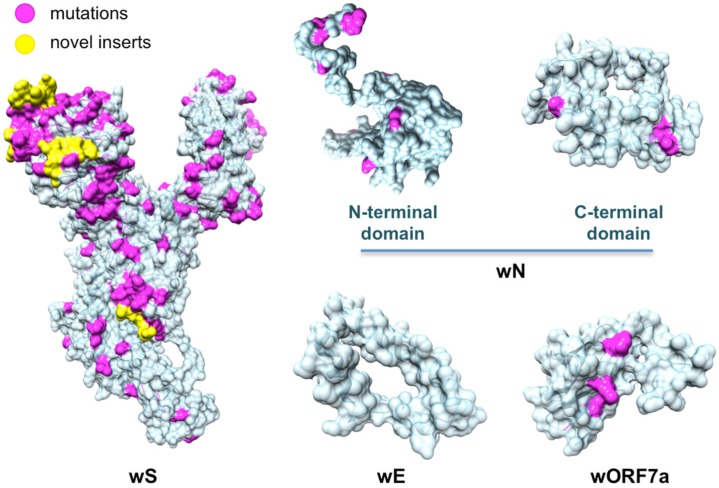
Structurally characterized structural proteins and an ORF of SARS-CoV-2. Highlighted in pink are mutations found when aligning the proteins against their homologs from the closest related coronaviruses: human SARS-CoV, bat coronavirus BtCoV, and another bat betacoronavirus BtRf-BetaCoV. Highlighted in yellow are novel protein inserts found in wS.

**Figure 6 viruses-12-00360-f006:**
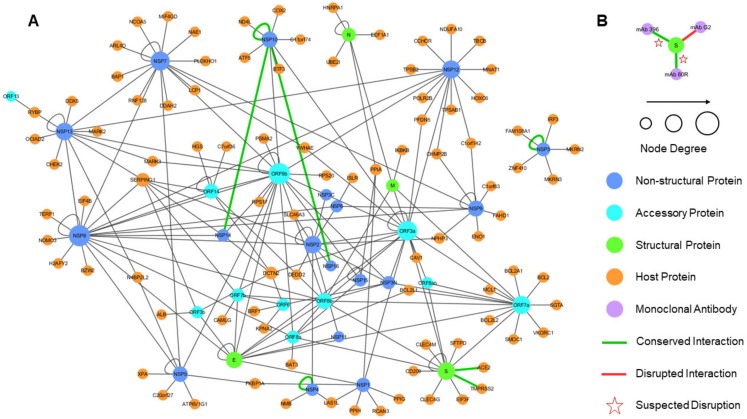
The unified interactome with SARS-CoV-2 interactions inferred through homology. (**A**) The SARS-CoV intra-viral and virus–host interactions are merged to create a unified interaction network indicating the types of proteins; the size of a node reflects the node degree. The SARS-CoV-2 interactions that are inferred from SARS-CoV are represented by green edges, with stars indicating the suspected disruption of the interaction based on the evolutionary conservation of the binding sites. The loops correspond to homo-oligomeric interactions. (**B**) Structural modeling prediction of interaction between SARS-CoV-2 S protein and three monoclonal antibodies.

**Table 1 viruses-12-00360-t001:** The list of SARS-CoV-2 proteins analyzed and structurally characterized in this work.

Protein	Accession	wORF1ab Region	Modeled Length	Template PDB id	Trgt-Tmplt Seq ID	Organism
wS, surface glycoprotein	YP_009724390		1273	6ACK	75%	SARS-CoV
wE, envelope protein	YP_009724392		75	5X29	89%	SARS-CoV
wORF7a	YP_009724395		121	1YO4	90%	SARS-CoV
wN, nucleocapsid phosphoprotein	YP_009724397		419	2JW8	96%	SARS-CoV
				1SSK	83%	SARS-CoV
				4UD1	51%	MERS-CoV
wNsp1	YP_009725297	13-127	115	2HSX	86%	SARS-CoV
wNsp3-domain1	YP_009725299	819-926	107	2GRI	79%	SARS-CoV
wNsp3-domain2	YP_009725299	1024-1198	175	2ACF	72%	SARS-CoV
wNsp3-domain3	YP_009725299	1232-1494	263	2WCT	76%	SARS-CoV
wNsp3-domain4	YP_009725299	1495-1550	66	2KAF	70%	SARS-CoV
wNsp3-domain5	YP_009725299	1564-1878	315	3E9S	82%	SARS-CoV
wNsp3-domain6	YP_009725299	1908-2763	113	2K87	82%	SARS-CoV
wNsp4	YP_009725300	3173-3263	91	3VC8	60%	MHV
wNsp5	YP_009725301	3264-3569	306	2GT7	96%	SARS-CoV
wNsp7	YP_009725302	3860-3942	83	1YSY	67%	SARS-CoV
wNsp8	YP_009725304	4019-4132	114	6NUR	85%	SARS-CoV
wNsp9	YP_009725305	4041-4253	113	3EE7	99%	SARS-CoV
wNsp10	YP_009725306	4262-4382	121	2G9T	98%	SARS-CoV
wNsp12	YP_009725307	4542-5311	770	6NUR	97%	SARS-CoV
wNsp13	YP_009725308	5325-5920	596	6JYT	100%	SARS-CoV
wNsp14	YP_009725309	5926-6451	526	5C8U	95%	SARS-CoV
wNsp15	YP_009725310	6452-6797	346	2H85	86%	SARS-CoV
wNsp16	YP_009725311	6800-7087	288	2XYQ	94%	SARS-CoV

**Table 2 viruses-12-00360-t002:** Network parameters of the SARS-CoV intra-viral, virus–host and unified networks. The table shows topological statistics for the three networks. Among the many computed statistics, the shown parameters include the number of nodes and edges in the networks, the average degree, number of components (independent networks), diameter (maximum shortest path), and clustering coefficient.

Network Parameters	SARS-CoV Intra-viral Interactome	SARS-CoV-Host Interactome	Unified Interactome
No. of nodes	31	118	125
No. of edges	86	114	206
No. of components	1	8	2
Diameter	4	14	7
Average degree	4.710	1.95	3.04
Clustering coefficient	0.448	0.0	0.068

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
