# Peer review of "Structural Genomics of SARS-CoV-2 Indicates Evolutionary Conserved Functional Regions of Viral Proteins"

_viruses, 2020, doi:10.3390/v12040360_

Round 1

Reviewer 1 Report

This is an extremely important and timely study that is crucially needed to the field. Conducted research is at the highest level and the generated data are of vital importance. Many researchers will find this work useful for their studies.  

Author Response

We sincerely thank this Reviewer for providing their feedback in such a short time.

Reviewer 2 Report

In this manuscript, the authors used computational methods to predict the structures of several SARS-CoV-2 proteins and then further predict their potential interactions. They also identified specific genomic changes that have occurred from SARS-CoV-2 to its recent relatives.

While I believe this is a unique and important piece of work, it clearly was rushed unnecessarily to publication. Aside from grammatical problems found throughout, there were a number of mistakes made with regard to figure numbers, etc. As a result, the manuscript was very difficult to read. Below I will outline some ways to clear up the article.

Major Issues:

  1. The writing was poor. There were many issues with past vs present tense. For instance, in lines 20 and 21 leverage and integrate should be in past tense. The methods section should also all be in past tense.
    1. Lines 43-44. This was nearly unreadable.
    2. Line 72. Don’t refer to the virus as the Wuhan SARS-CoV-2, it’s just SARS-CoV-2.
    3. When referring to the virus, SARS and MERS should always be written as SARS-CoV and MERS-CoV to not mix up the virus with the disease.
    4. Line 148. Should be wNsp3, not wNsp1
    5. Line 215. Did you mean Fig. 5D?
  1. The supplemental files are hard to read. Many times the authors say “see Suppl. Materials”. Instead they should cite a specific file. Second point, the alignments don’t list the specific virus and instead have a list of nearly 15 random sequences. I cannot easily tell which protein is from what virus. What does SP and TR mean. These should be simplified so that readers can easily determine what is going on.
  1. For table 1, it would be helpful to have another column showing the amino acids that encompass the different non-structural proteins and the protein domains of nsp3. Also, what residues in nsp3 constitute the 25-41 inserted residues in between domain 2 and domain 1 of nsp3? The ADRP domain is now called Macrodomain-1 or Mac-1 (Neuman B.W. 2016).

Minor Issues:

  1. The title is a little too long. I would go with “Structural genomic of SARS-CoV-2 indicate evolutionary conserved functional regions of viral proteins”. I think the conserved functional regions can imply interactions so there’s no need to state the interactomics part, especially since its just predicted.
  2. Is there really a coronavirus just called “Bat Coronavirus”? I find this hard to believe. This virus should have a more distinguishable name.
  3. Lines 130-137. Does the 2013 isolate really not have Orf10? I wonder if the nucleotide insertion could have been a sequencing mistake or that they sequenced a quasispecies that just happened to have a mix of sequences. The fact that one nucleotide difference would make a full-length protein makes me a little suspicious about the sequencing.
  4. Lines 172-173. I believe the figure being referenced is Fig. 1, not Fig. 3. Second, I counted 13 homo-oligomeric complexes and 6 human-virus interactions.
  5. Line 278. Are you really referring to Fig. 1 here? I didn’t see any green edges on Fig. 1.
  6. Figure 6. What are the loops around each center protein?

I would suggest going through the manuscript very closely for other potential mistakes. I likely did not find all of them.

Author Response

We sincerely thank this Reviewer for providing useful comments to improve the manuscript’s quality. We apologize about the poor writing: it was indeed a somewhat new experience for our group to complete this study and writing it up in such a short term. We have worked on the manuscript’s clarity and readability and hopefully improved the writing.

Major Issues:

The writing was poor. There were many issues with past vs present tense. For instance, in lines 20 and 21 leverage and integrate should be in past tense.

It has been fixed. The past tense in Results and Methods was made more consistent.

The methods section should also all be in past tense.

In writing the Methodology in the present tense, we were following the usual guidelines of  bioinformatics and computational biology journals, where the Results are written in the past tense, while the Methods are written in the present tense. We have modified our Methods to be in the past tense.

Lines 43-44. This was nearly unreadable.

We have rephrased that sentence.

Line 72. Don’t refer to the virus as the Wuhan SARS-CoV-2, it’s just SARS-CoV-2.

It has been made consistent

When referring to the virus, SARS and MERS should always be written as SARS-CoV and MERS-CoV to not mix up the virus with the disease.

We have made it consistent.

Line 148. Should be wNsp3, not wNsp1

It has been fixed.

Line 215. Did you mean Fig. 5D?

We meant Fig. 5 (all panels). It has been fixed.

The supplemental files are hard to read. Many times the authors say “see Suppl. Materials”. Instead they should cite a specific file.

We have now included references to specific file in Supplementary Materials.

Second point, the alignments don’t list the specific virus and instead have a list of nearly 15 random sequences. I cannot easily tell which protein is from what virus. What does SP and TR mean. These should be simplified so that readers can easily determine what is going on.

This is a default expanded UniProt entry format that is typically used when these alignments are calculated. We have now included a legend in each file that has abbreviation used for different coronavirus and their strains/isolates

For table 1, it would be helpful to have another column showing the amino acids that encompass the different non-structural proteins and the protein domains of nsp3.

We have included the regions for all proteins/domains of the polyprotein that have been modeled, including the domains of nsp3

Also, what residues in nsp3 constitute the 25-41 inserted residues in between domain 2 and domain 1 of nsp3?

It is region 983-1023. We have included that into description.

The ADRP domain is now called Macrodomain-1 or Mac-1 (Neuman B.W. 2016).

It has been corrected.

Minor Issues:

The title is a little too long. I would go with “Structural genomic of SARS-CoV-2 indicate evolutionary conserved functional regions of viral proteins”. I think the conserved functional regions can imply interactions so there’s no need to state the interactomics part, especially since its just predicted.

We thank the Reviewer for suggestion. We like the proposed title and have adopted it here.

Is there really a coronavirus just called “Bat Coronavirus”? I find this hard to believe. This virus should have a more distinguishable name.

According to both, NCBI and UniProt, the species are classified either as “Bat coronavirus” (different isolates) or Bat Coronavirus, BtCov . See examples:

https://www.ncbi.nlm.nih.gov/nuccore/KU182964

https://www.ncbi.nlm.nih.gov/nuccore/DQ648857.1/

To mitigate this concern, we have included the strain/isolate and year of collection information into Table S1.

Lines 130-137. Does the 2013 isolate really not have Orf10? I wonder if the nucleotide insertion could have been a sequencing mistake or that they sequenced a quasispecies that just happened to have a mix of sequences. The fact that one nucleotide difference would make a full-length protein makes me a little suspicious about the sequencing.

You are right. Upon checking, it looks like it was a sequencing error indeed. The error has been fixed now and this region contains the full Orf10 identical to that one of  SARS-CoV-2. We have removed that paragraph.

Lines 172-173. I believe the figure being referenced is Fig. 1, not Fig. 3.

It should have been Figs 1 and 4. It has been fixed.

Second, I counted 13 homo-oligomeric complexes and 6 human-virus interactions.

These numbers include multiple conformations modeled. We have clarified that.

Line 278. Are you really referring to Fig. 1 here? I didn’t see any green edges on Fig. 1.

It should have been Fig. 6. It has been fixed.

Figure 6. What are the loops around each center protein?

They correspond to homo-oligomers (a protein interacting with another copy of it). We have inserted this annotation into Fig. 6 legend.

I would suggest going through the manuscript very closely for other potential mistakes. I likely did not find all of them.

We have gone through the manuscript and indeed found several other typos and places that needed clarification. Thank you.

Round 2

Reviewer 2 Report

This manuscript has been significantly improved since its first submission. There are only a few minor grammatical mistakes to change.

1. I have to apologize, the title I proposed was not grammatically correct, it should be either "Structural genomics ...." or "Structural genomic analysis...".

2. Lines 14-15. "over one-hundred thousand people worldwide and has taken.."

3. Line 20. "We leveraged data on related coronaviruses"

4. Line 25. There is an extra space before the period.

5. Line 28. remove Wuhan

6. Line 39. "two and a half months since its initial ....."

7. Line 87. Surface should be Spike

8. Line 148. Should be Mac-1

9. Line 278 and Fig. 6. There is no suspected disruption between S and ACE2. It is well-established now that ACE2 is the receptor for SARS-CoV-2 and binds at least as well if not better to the SARS-2 spike protein than the SARS-CoV spike protein.

Author Response

1. I have to apologize, the title I proposed was not grammatically correct, it should be either "Structural genomics ...." or "Structural genomic analysis...".

Thank you for the note. We corrected it. We also believe that it should be "indicates" not "indicate". So we corrected that as well.

2. Lines 14-15. "over one-hundred thousand people worldwide and has taken.."

Fixed.

3. Line 20. "We leveraged data on related coronaviruses"

Fixed.

4. Line 25. There is an extra space before the period.

Fixed that and several other extra space issues.

5. Line 28. remove Wuhan

Fixed.

6. Line 39. "two and a half months since its initial ....."

Fixed.

7. Line 87. Surface should be Spike

We find that "surface glycoprotein" is often used as well as "spike surface", so we modified it to "Surface glycoprotein (or Spike)"

8. Line 148. Should be Mac-1

Fixed

9. Line 278 and Fig. 6. There is no suspected disruption between S and ACE2. It is well-established now that ACE2 is the receptor for SARS-CoV-2 and binds at least as well if not better to the SARS-2 spike protein than the SARS-CoV spike protein.

Thank you for noticing it. We have also read about this surprising fact (due to a high number of mutations in the S-ACE2 interaction interface), but forgot to correct our speculations. We have now modified it to "Surprisingly, it was recently established that the interaction was preserved [65], in spite of a number of mutations in the receptor binding site of wS. "